# Late-stage synthesis of heterobifunctional molecules for PROTAC applications via ruthenium-catalysed C–H amidation

Daniele Antermite [1], Stig D. Friis [1], Johan R. Johansson [1], Okky Dwichandra Putra [2], Lutz Ackermann [3,4] ✉ & Magnus J. Johansson [1] ✉

PROteolysis TArgeting Chimeras (PROTACs) are heterobifunctional molecules emerging as a powerful modality in drug discovery, with the potential to address outstanding medical challenges. However, the synthetic feasibility of PROTACs, and the empiric and complex nature of their structure-activity relationships continue to present formidable limitations. As such, modular and reliable approaches to streamline the synthesis of these derivatives are highly desirable. Here, we describe a robust ruthenium-catalysed late-stage C–H amidation strategy, to access fully elaborated heterobifunctional compounds. Using readily available dioxazolone reagents, a broad range of inherently present functional groups can guide the C–H amidation on complex bioactive molecules. High selectivity and functional group tolerance enable the late-stage installation of linkers bearing orthogonal functional handles for downstream elaboration. Finally, the single-step synthesis of both CRBN and biotin conjugates is demonstrated, showcasing the potential of this methodology to provide efficient and sustainable access to advanced therapeutics and chemical biology tools.

Over the last 20 years, PROteolysis TArgeting Chimeras (PROTACs) have emerged as a new exciting class of therapeutic agents with the potential to tackle previously inaccessible pharmacological targets[1–3]. PROTACs are heterobifunctional molecules consisting of a ligand for a target protein of interest (POI) and a ligand for an E3-ubiquitin ligase (E3), such as Cereblon (CRBN)[4] or von Hippel-Lindau (VHL)[5], connected by a suitable linker (Fig. 1a)[6,7]. This molecular architecture allows PROTACs to bind simultaneously to a POI and an E3, forming a ternary complex which in turn leads to selective POI degradation[8]. This event-driven mode of action offers the intriguing potential to modulate traditionally undruggable targets, while overcoming other shortcomings of typical small-molecule drugs including off-target and resistance mechanisms[9,10]. As such, the PROTAC modality has sparked tremendous interest both in

academia and the pharmaceutical industry, with several PROTACs currently undergoing clinical trials[3].

Formation of a productive ternary complex requires an optimal combination of all PROTAC components, resulting in highly complex structure-activity relationships (SAR)[11]. In addition to binary binding efficiency, the type and length of the linker and its attachment points to the POI- and E3-ligands can dramatically affect potency and physicochemical properties of the final PROTAC[12]. Despite major advances in structural biology[8,13] and computational modelling[9,14], the rational conversion of a POI binder into a functional degrader, and its SAR exploration, remain primarily empirical and based on many iterations of design-make-test-analyse (DMTA) cycles, largely driven by synthetic feasibility[6]. This results in time- and resource-intensive synthetic efforts posing a significant burden

[1]Medicinal Chemistry, Research and Early Development, Cardiovascular, Renal and Metabolism (CVRM), BioPharmaceuticals R&D, AstraZeneca, Gothenburg, Sweden. [2]Early Product Development and Manufacturing, Pharmaceutical Sciences R&D, AstraZeneca, Gothenburg, Sweden. [3]Institut für Organische und Biomolekulare Chemie, Georg-August-Universität Göttingen, Göttingen, Germany. [4]German Center for Cardiovascular Research (DZHK), Berlin, Germany. ✉e-mail: lutz.ackermann@chemie.uni-goettingen.de; magnus.j.johansson2@astrazeneca.com

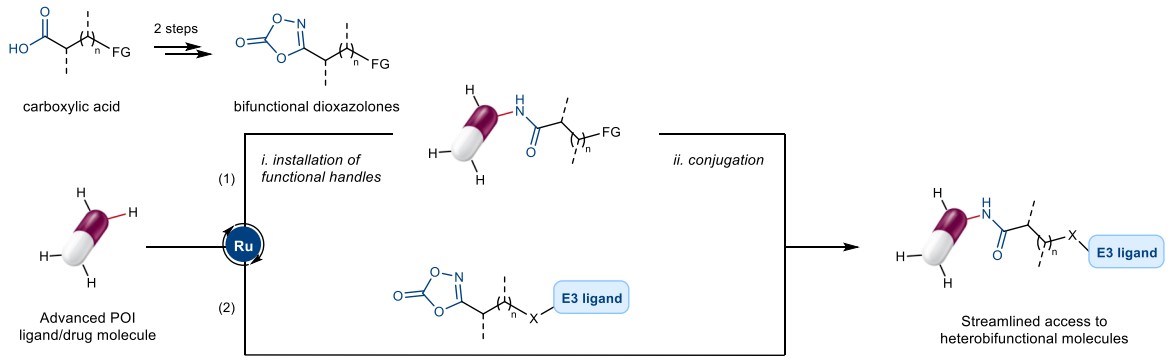

**a  Examples of PROteolysis TArgeting Chimeras (PROTACs) in clinical trials & key synthetic steps**

**ARV-110 -** AR Degrader

**ARV-471 -** ER Degrader

**b  This work: Synthesis of PROTAC-like compounds via late-stage C–H amidation**

*single-step reaction with E3-ligand bound dioxazolones*

**Fig. 1 | Developing a late-stage C–H functionalisation platform applicable to the synthesis of heterobifunctional compounds. a** PROTACs consists of 3 different structural components: a POI-ligand, an E3-ligand, and a linker. As representative examples, the chemical structures of ARV-110 and ARV-471, the first 2 PROTACs to enter clinical trials, are shown. Multistep and labour-intensive synthetic sequences are required in PROTAC discovery, typically relying on the de novo synthesis of pre-functionalised POI-ligand precursors. **b** Ruthenium-catalysed late-stage C–H amidation with readily available dioxazolone reagents. Streamlined access to PROTAC-like molecules and other drug conjugates is provided through direct C–H functionalisation of advanced POI-ligands, either in a stepwise (i.e. installation of functional handles for subsequent conjugation, path 1, or single-step approach, path 2). POI protein of interest, CRBN cereblon E3 ligase, AR androgen receptor, ER estrogen receptor, FG functional handle (for conjugation), X linker attachment.

to PROTAC discovery programs. Traditional strategies such as amide bond formation, reductive amination, nucleophilic aromatic substitution and *N*-alkylation[7], have recently been complemented by solid-phase synthesis[15], click-chemistry[16], Staudinger ligation[17] and multicomponent reactions[18] in order to improve synthetic throughput. Direct-to-biology and miniaturised approaches have also been developed to speed up the DMTA cycle[19,20]. However, all these strategies rely on the pre-functionalisation of the POI-ligand with a suitable functional handle along the desired exit vector(s). This itself often represents a considerable synthetic hurdle, and modular and reliable approaches to streamline the synthesis of PROTACs continue to be in high demand.

In recent years, late-stage functionalisation (LSF) has taken huge strides emerging as a powerful approach for the assembly and diversification of increasingly complex molecules[21–23]. Exiting advances in transition metal-catalysed C–H functionalisation have provided access to new disconnections with increased atom- and step-economy, expanding considerably the medicinal chemistry toolbox[24–26]. Powerful strategies have been described to enable the installation of reactive handles for elaboration of complex scaffolds[27,28], and the introduction of small substituents with potential to influence profoundly the physicochemical properties of drug compounds[29–31]. However, issues of site-selectivity and functional group compatibility continue to limit the development of new late-stage C–H disconnections[32]. Furthermore, methods that allow direct conjugation of two complex molecular entities are largely absent in the LSF landscape, despite the huge significance that such an approach would offer for the construction of advanced drug conjugates.

In order to expedite access to PROTACs, we envisaged a robust C–H functionalisation method that could be generally applicable to complex bioactive molecules in a late-stage fashion, thus avoiding the need for lengthy and tedious de novo synthesis of pre-functionalised POI-ligands (Fig. 1b). To achieve high site-selectivity without relying on installation and removal of specialised directing groups[33], we aimed at taking advantage of functional groups frequently found in drug molecules to guide the C–H activation through Lewis basic coordination to the metal catalyst. At the same time, exceptional tolerance to polar functionalities was recognised as a key requirement of the transformation to accommodate substrate, linker and E3-ligand components. To this end, we report herein a late-stage ruthenium-catalysed C–H amidation platform using diversely functionalised dioxazolone coupling partners. These were identified as desirable amidating reagents that can leverage extensive libraries of commercially available carboxylic acid precursors, while avoiding the hazards and sluggish reactivity of other nitrogen sources[34–42]. Featuring a broad scope of inherent directing groups, this strategy enabled the late-stage installation of a wide range of functional handles for downstream elaboration, as well as the single-step synthesis of both CRBN and biotin heterobifunctional conjugates from complex drug molecules.

## Results and discussion
### Multiparameter optimisation
At the outset, a high-throughput experimentation (HTE) campaign was deemed as the most time- and resource-efficient strategy to identify optimal reaction conditions that could be applicable across

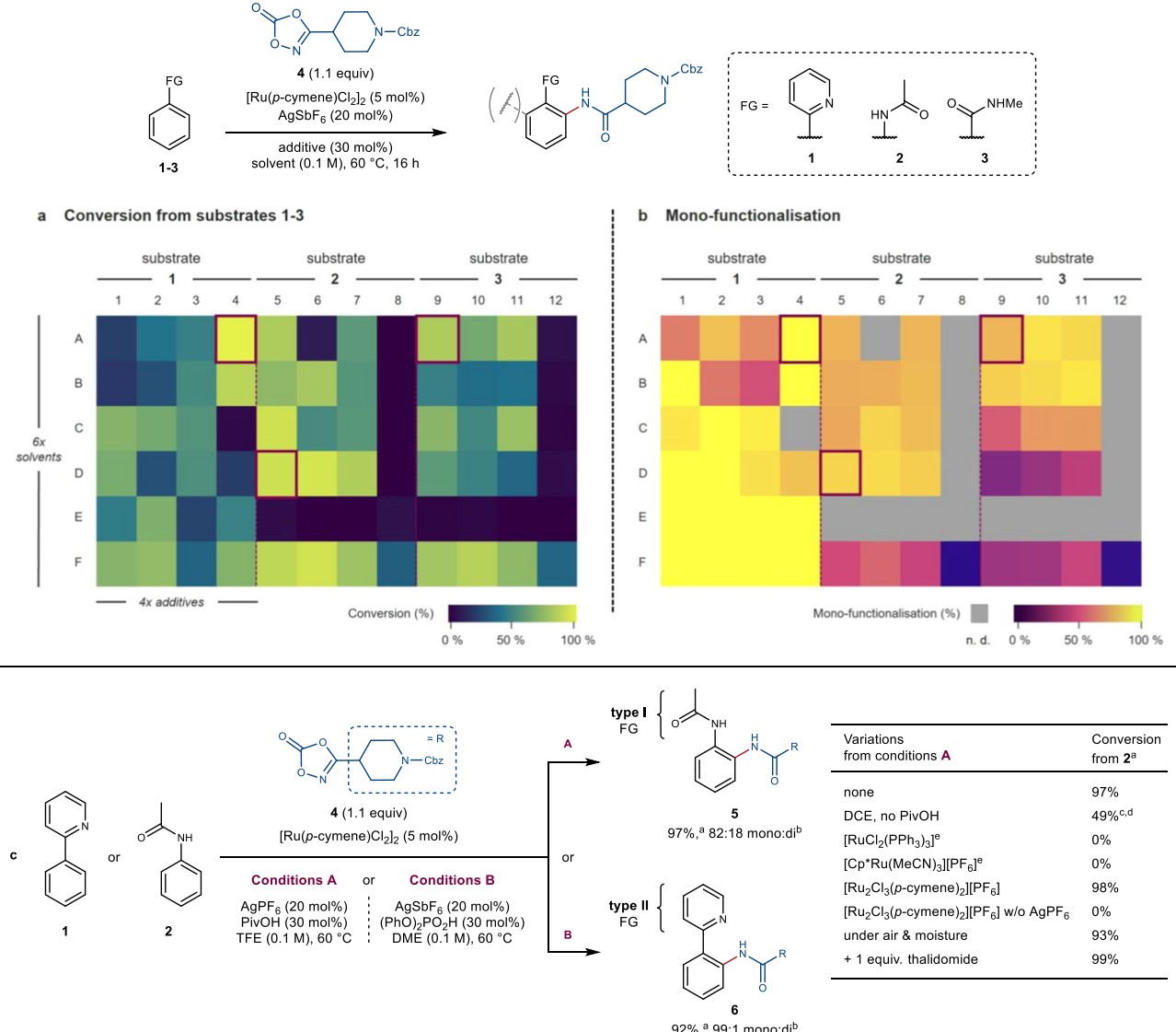

**Fig. 2 | High-throughput multiparameter optimisation of ruthenium-catalysed C–H amidation with dioxazolones. a**, **b** Selective cross-optimisation of solvents and additives for the C–H amidation of substrates **1-3** visualising conversion **a** and levels of mono-functionalisation **b**. Best conditions for each substrate are highlighted in purple boxes (0.25 μmol scale). n. d. Not determined due to low conversion (<10%). Solvents (rows): DME, 1,2-dimethoxyethane (A); EtOAc (B); DCE, 1,2-dichloroethane (C); TFE, 2,2,2-trifluoroethanol (D); HFIP, 1,1,1,3,3,3-hexafluoro-2-propanol (E); TFT, α,α,α-trifluorotoluene (F). Additives (columns): PivOH, pivalic acid (1,5,9); 2,4,6-trimethylbenzoic acid (2,6,10); N-acetyl glycine (3,7,11); (PhO)$_2$PO$_2$H (4,8,12). **c** Optimised conditions for type I (**5**) and II (**6**) directing functional groups (0.25 mmol scale). [a] Total conversion (%) determined by LC-MS. [b] Ratio of mono- vs bis-functionalised product determined by LC-MS. [c] Preferential formation of rearranged carbamoylated product **15** (7:3) was observed (see Supplementary Information, Section 4). [d] 0.25 μmol scale. [e] 10 mol% [Ru]. FG directing functional group, Cbz benzyloxycarbonyl.

challenging pharmaceutically relevant chemical space[43,44]. Preliminary investigations into rhodium-, cobalt- and ruthenium-based catalytic systems revealed the latter to offer high potential for the development of such a C–H amidation platform. A selective multiparameter optimisation was thus undertaken for the ruthenium-catalysed reaction of dioxazolone **4** with compounds **1-3** (Fig. 2). These were chosen to represent different levels of Lewis basicity and potential coordination modes to the ruthenium centre, which were anticipated to affect both the success of reaction and the tendency towards poly-functionalisation[45]. Discrete and continuous parameters likely to show interdependence were subjected to cross-optimisation to maximise both conversion and mono-functionalisation on each substrate (see Supplementary Information, Section 4). PROTAC relevant functionalities, such as those present in the CRBN-binder thalidomide, are often poorly tolerated in metal-catalysed cross-couplings[7]. Therefore, the efficiency of the reaction in the presence of thalidomide as an additive was also used as a key robustness parameter to select reaction conditions throughout the optimisation process.

Intriguingly, the combination of solvent and acidic additive was found to profoundly influence the reaction outcome on the different substrates (Fig. 2a, b). While higher levels of mono-functionalisation were generally observed for 2-phenyl pyridine **1**, the use of 1,2-dimethoxyethane (DME) as solvent in combination with 30 mol% (PhO)$_2$PO$_2$H was crucial to achieve optimal reactivity (well A4). In particular, the phosphate additive significantly improved conversion by suppressing detrimental homocoupling of **1**. In contrast, these conditions failed to provide any product in the reaction of **2** and **3**. Curtius rearrangement leading to C–H carboxyamidation of acetanilide **2** was observed in the absence of any additive (see Supplementary Information, Section 4). However, this could be suppressed by addition of a carboxylic acid to facilitate the desired reaction pathway. Pivalic acid (PivOH) was an optimal additive in the reaction of

both **2** and **3**, while the use of 2,2,2-trifluoroethanol (TFE) as solvent was key to minimise bis-functionalisation of more electron-rich anilide **2** (well D5). Further optimisation studies finally led to two sets of reaction conditions (A and B), both employing [Ru(*p*-cymene)Cl$_2$]$_2$ as pre-catalyst (Fig. 2c). These provided optimal results respectively with more weakly (type I) and more strongly (type II) Lewis basic functional groups. Dimeric cationic ruthenium complex [Ru$_2$Cl$_3$(*p*-cymene)$_2$] [PF$_6$] was also a competent ruthenium source, though still requiring catalytic silver salt. These mild conditions were not affected by moisture and air, and most importantly were compatible with the thalidomide additive used as benchmark of functional group tolerance.

## LSF informer library

Having designed and optimised these reaction conditions to tolerate complex functionalities of medicinal relevance, we set out to assess the generality and robustness of our protocol more broadly. To this end, we used an informer library approach[46,47], whereby 48 commercial drugs were evaluated in the reaction with dioxazolone **4** (Fig. 3a). These were selected to present at least one suitable C(sp$^2$)–H amidation site, as well as to display high structural diversity in order to sample broad and pharmaceutically relevant chemical space (Fig. 3b). Only compounds bearing primary or secondary nucleophilic amines were excluded from the library due to incompatibility with the dioxazolone reagent[48]. We reasoned that this approach would provide valuable information, probing both the site-selectivity in the presence of multiple susceptible C–H bonds, and the effect of different inherent directing groups on the reaction outcome. Furthermore, this was anticipated to allow a rapid assessment of the compatibility of the reaction with different functional groups typically found in modern active pharmaceutical ingredients (APIs)[49].

Upon preliminary structural evaluation, the different LSF substrates were clustered into two 96-well plates based on the type (I or II) of functional group(s) available to direct the C–H activation. Three sets of reaction conditions, including general conditions A and B as well as variations thereof, were screened for each compound for a total of 144 reactions (see the Supplementary Information, Section 5). A 9:1 DME:DMA solvent mixture was also evaluated to account for the poor solubility expected for some of the complex LSF substrates. Pleasingly, a 46% success rate – defined as the number of substrates giving > 30% conversion to the amidated product – was obtained, with 15 dugs resulting in > 60% conversion. As expected from the optimisation studies, the combination of TFE and PivOH gave better results in the presence of type I inherent directing groups, while DME/(PhO)$_2$PO$_2$H continued to be a superior set of reaction conditions for substrates containing more strongly Lewis basic functionalities. Importantly, the formation of single products was observed in most cases, highlighting high levels of mono-functionalisation as well as of regio- and chemo-selectivity across the different molecular scaffolds. A variety of unprotected functionalities were successfully tolerated, including carboxylic acids, alcohols and sulphonamides (vide infra). In contrast, reduced reactivity was observed in the presence of tertiary amines and other mono- or bidentate moieties that could result in strong unproductive coordination to the ruthenium centre (e.g. sonidegib, apixaban and minaprine, Fig. 3b).

Analysis of the successful examples revealed that 15 different classes of functional groups, inherently part of the drug scaffolds, could be leveraged to direct the late-stage C–H amidation (Fig. 3c). This included some of the most ubiquitous functionalities in natural products and pharmaceuticals, such as secondary benzamides and anilides. Ketones were also productive directing groups despite their weaker Lewis basic character. Furthermore, nitrogen-containing imines and a variety of azoles and azines displayed good to excellent reactivity. Interestingly, this reaction could also be promoted by 2-aryloxy and 2-aminopyridine derivatives, which have not been

previously reported as competent directing groups in C–H amidation, likely via formation of 6-membered ruthenacycle intermediates (Fig. 3c, extended azines).

## Scope of late-stage C–H amidation

To corroborate these promising results, the scope of this late-stage C–H amidation was then explored on synthetically useful scale. The reaction could be scaled up 10 times from the HTE conditions (0.25 mmol and 25 µmol respectively), with no reduction in efficiency or selectivity. Using 5 mol% [Ru(*p*-cymene)Cl$_2$]$_2$ and a minimal excess of dioxazolone **4**, 22 marketed drugs were successfully amidated with high chemo- and regioselectivity (Fig. 4). Sulfaphenazole derivative **8a** was formed in nearly quantitative yield, with no side-reactivity arising from either the unprotected aniline or sulphonamide moieties. This outstanding robustness was a general feature of the transformation, allowing its application to highly complex and densely functionalised medicinal compounds. Furthermore, this led to clean reaction profiles across the scope, with unreacted starting material typically accounting for the remaining mass balance (for LC-MS traces of the crude reaction mixtures see Supplementary Information, Sections 9.3–9.5), which in turn enables the high-value substrates to be easily recovered upon purification. Competing acylation of the primary sulphonamide in celecoxib was observed under conditions B. However, more protic conditions A allowed exclusive formation of product **8b** in 74% yield, highlighting the impact of our initial HTE studies to identify multiple viable reaction conditions. Other azole-containing compounds were also successful, giving **8c**–**8e** in moderate to excellent yields while being compatible with common acidic groups like carboxylic acids and acyl sulphonamides. The peptide-like backbone of atazanavir and the thiazolylamide scaffold of pritelivir were well tolerated, resulting in mono-amidated products **8f** and **8g** in 66% and 64% yield, respectively. Similarly, conditions B successfully promoted the late-stage amidation in the presence of pyrimidine (**8h**), pyrazine (**8i**) and imine (**8j**) innate directing groups. Notably, diflufenican (**8k**) and talniflumate (**8l**) underwent selective C–H amidation *ortho*- to the 2-aryloxy and 2-aminopyridine functionalities, with no acetal deprotection observed for prodrug talniflumate. Alkyl, aryl and alkenyl anilides all proved to be excellent substrates even when more sterically encumbered, facilitating C–H activation on a variety of different compounds (**8m**–**8s**). Excellent regioselectivity for the most electron-rich and sterically accessible C(sp$^2$)–H bond was achieved in all cases, despite the presence of multiple potentially reactive sites. This was exemplified by tolvaptan, whereby only 1 of 5 possible C(sp$^2$)–H bonds was selectively functionalised in 97% yield (**8r**), as confirmed by X-ray crystallography. Examples of ruthenium-catalysed C–H functionalisation of benzanilide derivatives have been reported both at the anilide and benzamide positions, with different selectivity depending on the specific reaction manifold and on the conformational flexibility of the substrate[50–52]. In our case, the remarkable site-selectivity observed for the anilide *ortho*-C–H bonds can likely be ascribed to both a conformational preference to form 6-membered ruthenacycle intermediates through carbonyl *O*-coordination[51–53], and a base-assisted electrophilic substitution-type (BIES) mechanism for the C−H activation step[54,55].

High chemoselectivity for the desired C–H functionalisation was observed in the reaction of aryl iodide trametinb (**8s**), as well as in the presence of phenols (**8o**) and aliphatic alcohols (**8p**–**8r**). Finally, benzamides (**8t**, **8u**) and ketones (**8v**–**8y**) could also be leveraged to guide the C–H amidation in the absence of other competing directing functionalities. Despite partial -OH acylation was observed for paclitaxel (22% yield, **28** in the Supplementary Information), it is remarkable that C–H amidation still occurred to form **8u** in 19% yield, enabling rapid access to analogues of this highly complex antineoplastic agent.

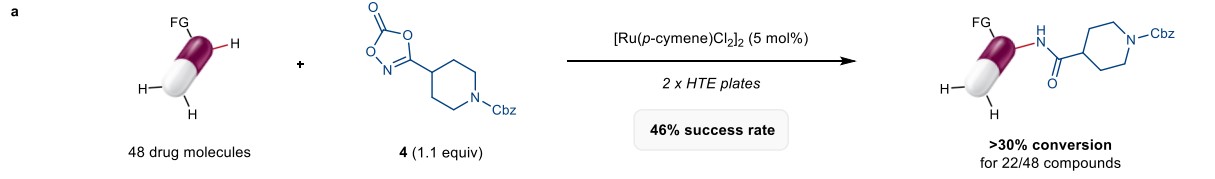

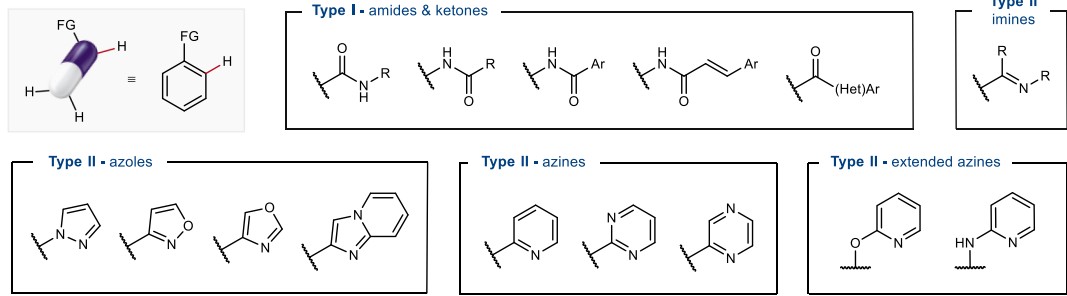

**Fig. 3 | LSF informer library to assess generality and robustness of the C–H amidation protocol. a** 48 marketed drugs were evaluated in the ruthenium-catalysed C–H amidation with dioxazolone **4** (0.25 µmol scale). Each compound was screened against three reaction conditions based either on optimal conditions A (cf. Figure 2c) or B (cf. Figure 2c). See the Supplementary Information, Section 5, for full details on library and reaction conditions. **b** Representative examples of successful and unsuccessful drugs in the informer library screening. The relative product abundance (%) was determined by LC-MS from the relative integration of product peak vs unreacted substrate and/or other by-products in the UV chromatogram. Among the various susceptible C–H bonds in each compound (highlighted in red), the site of C–H amidation is indicated in bold. Unreacted starting material represented the major component of the crude reaction mixture in the unsuccessful examples (see the Supplementary Information, Section 5 for substrate specific details). **c** 15 classes of inherent functional groups were found to enable late-stage C–H amidation on complex drug substrates. FG directing functional group, Cbz benzyloxycarbonyl, HTE high throughput experimentation, LSF late-stage functionalisation, R alkyl group, Ar aryl group, HetAr heteroaryl group.

Functionalisation of a POI ligand at a position that does not hinder binding to the target protein is of essential importance for the generation of successful PROTACs. Pleasingly, analysis of published X-ray structures of successful LSF substrates in complex with their protein targets revealed that C–H amidation can occur in a solvent exposed region of the molecule in a number of cases (**8c, 8f,** **8p,** and **8v**), and in the presence of both type I and II directing groups (see the Supplementary Information, Section 7). While no general conclusion can be drawn for all different substrates, this supports the potential use of this chemistry to access functional PROTACs through late-stage derivatisation of various POI ligands along suitable exit vectors.

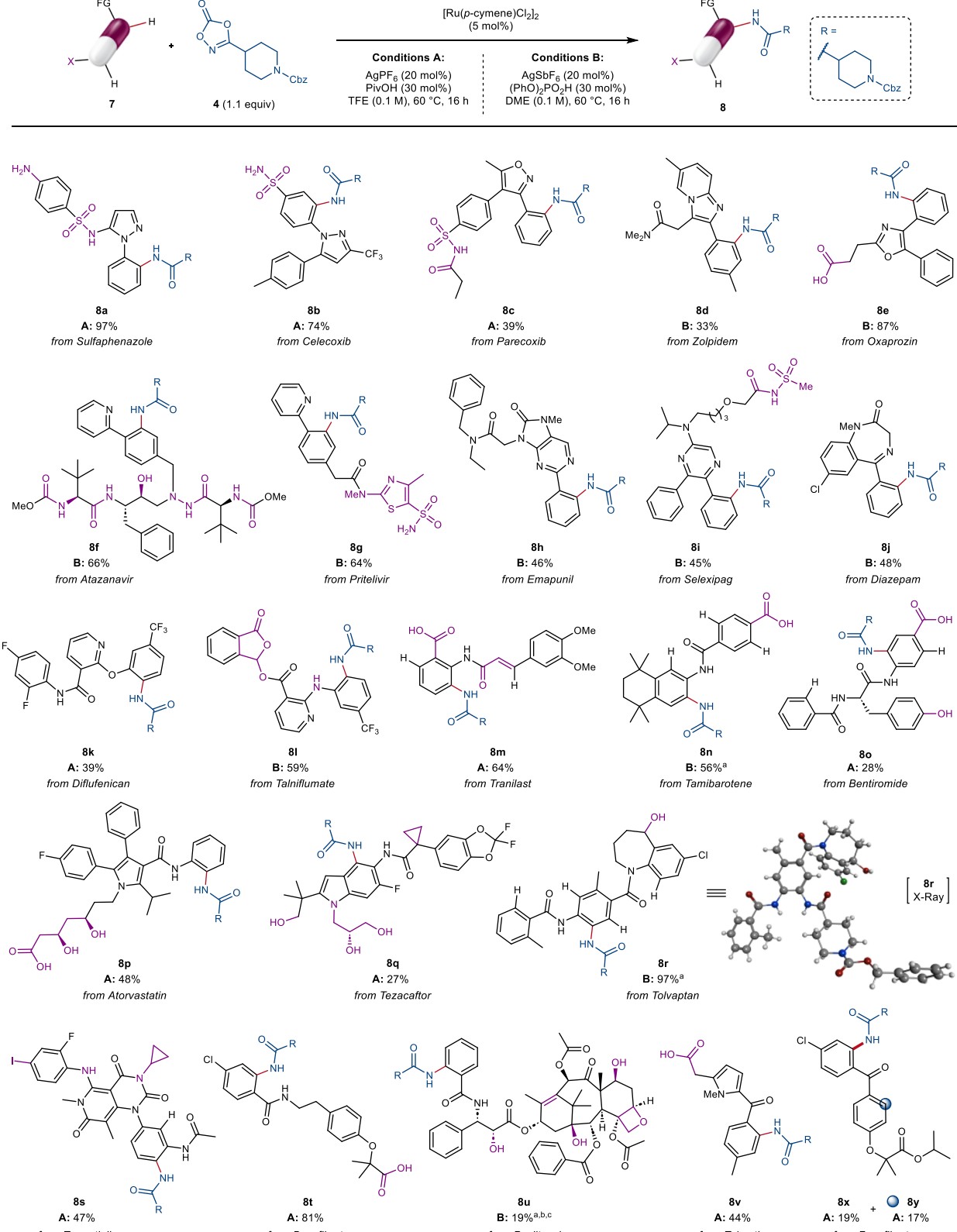

**Fig. 4 | Late-stage functionalisation of drug molecules via ruthenium-catalysed C–H amidation.** Yields refer to the isolated product (0.1–0.25 mmol scale). A broad range of polar, unprotected or sensitive functional groups (X, highlighted in purple) were tolerated, typically resulting in high mass recovery (see Supplementary Information, Section 9.3 for LC-MS traces of the crude reaction mixtures). [a] Pivalic acid (PivOH) was used in place of (PhO)₂PO₂H. [b] A mixture of DME:DMA, 9:1 was used as solvent. [c] Product **28**, derived from -OH acylation of paclitaxel was also isolated in 22% yield, while unreacted starting material accounted for the remaining mass balance (see Supplementary Information, Section 9.3). FG directing functional group, Cbz benzyloxycarbonyl, TFE 2,2,2-trifluoroethanol, DME 1,2-dimethoxyethane.

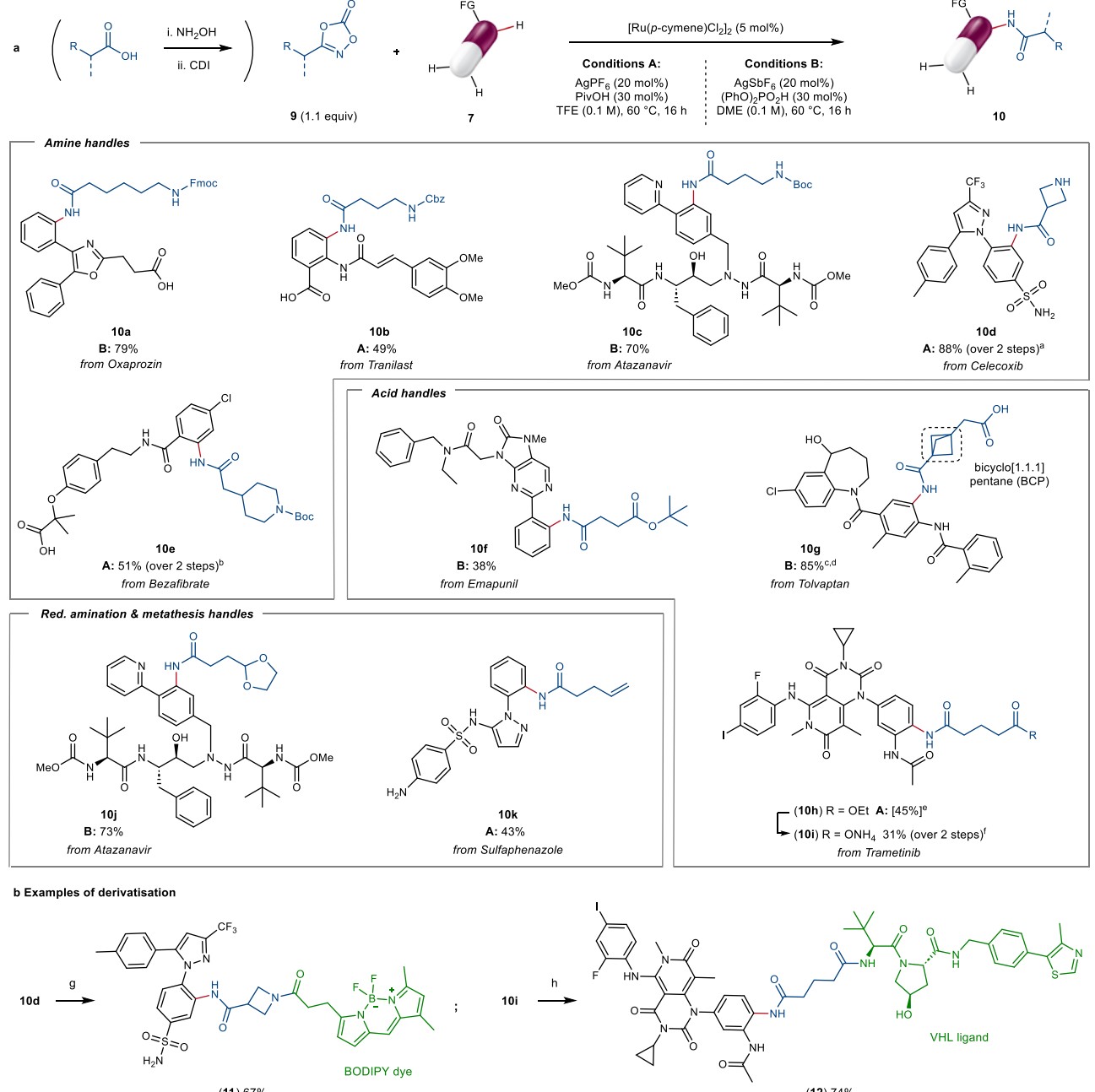

**Fig. 5 | Ruthenium-catalysed C–H amidation with bifunctional dioxazolones for late-stage linker installation. a** Late-stage installation of linkers offering orthogonal functional handles for subsequent derivatisation. Yields refer to the isolated product (0.1–0.25 mmol scale). [a] Partial Boc deprotection observed: **10d** was isolated as the free amine, after one-pot treatment with trifluoroacetic acid (TFA). [b] Partial Boc deprotection observed: **10e** was isolated after one-pot treatment with di-tert-butyl dicarbonate. [c] Pivalic acid (PivOH) used instead of (PhO)₂PO₂H. [d] In situ tert-butyl ester deprotection observed: **10g** was isolated as the free acid. [e] Conversion of substrate to **10h** after C–H amidation, as determined by LC-MS. [f] Carboxylate **10i** was isolated after telescoped saponification with NaOH. See the

Supplementary Information, Section 9.4 for substrate specific details. **b** Examples of derivatisation following late-stage C–H amidation: fluorescent tag labelling (**11**) and synthesis of VHL derivative **12**. [g] Reaction conditions: BDP-FL NHS ester (0.02 mmol), **10d** (1.2 equiv), DIPEA (1.2 equiv), MeCN, rt, 1 h. [h] Reaction conditions: **10i** (0.03 mmol), VHL ligand (1.1 equiv), HATU (1.2 equiv), DIPEA (3.0 equiv), DMF, rt, 2 h. CDI 1,1′-carbonyldiimidazole, FG directing functional group, TFE 2,2,2-trifluoroethanol, DME 1,2-dimethoxyethane, Fmoc fluorenylmethoxycarbonyl, Cbz benzyloxycarbonyl, Boc tert-butyloxycarbonyl, BODIPY boron-dipyrromethene, VHL von Hippel-Lindau E3 ligase.

Aiming to apply this methodology to the synthesis of PROTACs and other drug conjugates, we further investigated the performance of different dioxazolones in the reaction. In particular, we reasoned that the use of bifunctional reagents, bearing additional functional handles for downstream elaboration, would provide an exciting opportunity for the late-stage installation of diverse linkers into complex small molecule ligands, making this a very useful protocol for chemical biology applications (Fig. 5). To this end, carbamate-containing

dioxazolones proved to be excellent coupling partners, enabling the introduction of flexible amino-alkyl linkers (**10a–10c**), as well as of more rigid cyclic scaffolds (**10d, 10e**). A range of versatile amino protecting groups including Fmoc (**10a**), Cbz (**10b**), and Boc (**10c**) were tolerated. Importantly, this allows orthogonal deprotection conditions –e.g. basic, reductive or acidic – to be chosen based on the chemical stability of the different POI-ligand. Minor Boc deprotection was observed upon reaction of celecoxib and bezafibrate, likely due to

the Lewis acidic amidation conditions. Nonetheless, subsequent one-pot treatment of the crude mixture with trifluoroacetic acid afforded free NH azetidine **10d** in 88% yield. Alternatively, *N*-Boc protected derivative **10e** was exclusively isolated upon one-pot reaction with di-*tert*-butyl dicarbonate. Carboxylic acid derivatives were also successful, tolerating both acid- and base-labile ester functionalities (**10f**–**10i**), thus unlocking reverse amide-bond formation strategies that can take advantage of vast amine libraries. In recent years, the bicyclo[1.1.1] pentane (BCP) framework has attracted significant attention in drug design as a bioisosteric replacement for aromatic rings, resulting in superior pharmacokinetic properties such as improved solubility and metabolic stability[56]. Notably, this attractive motif could be successfully installed under our ruthenium-catalysed conditions, with no reduced reactivity deriving from increased steric hindrance around the dioxazolone. This afforded tolvaptan derivative **10g** in 85% yield as the free acid, after in situ *tert*-butyl ester deprotection. Similarly, C–H amidation followed by hydrolysis of the ethyl ester group yielded trametinib derivative **10i** directly as the corresponding carboxylate salt. Finally, the late-stage introduction of acetal-protected aldehyde (**10j**) and terminal olefin (**10k**) was demonstrated, paving the way for further conjugation approaches including reductive amination, meta-theses or nucleophilic substitution following hydroboration/ oxidation.

To showcase the synthetic utility of this strategy for linker installation, further derivatisation of the amidated LSF products was exemplified. In particular, fluorescent labelling of celecoxib derivative **10d** with a BODIPY (boron-dipyrromethene) dye was achieved in 67% yield (**11**, 59% over 3 steps from the parent drug). The von Hippel-Lindau (VHL) ligand was found incompatible with the C–H amidation conditions, likely due to poor tolerance towards the thiazole moiety (only traces of products **5** and **6** were found upon addition of 1 equiv of either *N*-Ac VHL ligand or 5-phenylthiazole to otherwise standard reaction conditions). Nonetheless, VHL-based derivatives could be easily accessed after linker attachment, as exemplified by product **12**. This was formed in 74% yield by simple amide coupling of trametinib-derived carboxylate **10i** (23% over 3 steps from the parent drug).

## Single-step conjugation

Encouraged by the high functional group tolerance demonstrated thus far, we became intrigued by the possibility to access fully elaborated PROTAC-like derivatives in a single step. As proof of concept, dioxazolones bearing various combinations of linkers and CRBN binders were synthesised and tested in the late-stage C–H amidation (Fig. 6a). To our delight, optimised conditions A and B afforded derivatives **14a** and **14b** in 67% and 42% yield respectively. It is remarkable that the reaction was not affected by the overall level of molecular complexity, with no detrimental effect from the multiple Lewis basic sites in substrate, ethylene glycol linker or pomalidomide warhead. The reaction proceeded smoothly in the presence of linear alkyl linkers with different attachment points to the pomalidomide core, as exemplified in the amidation of celecoxib (**14c**) and oxaprozin (**14d**). Further increasing complexity, installation of lenalidomide E3-ligand tethered to a BCP linker was demonstrated, giving atazanavir conjugate **14e** as the exclusive product. Importantly, type I directing groups were also successful in the single-step conjugation, affording tolvaptan derivative **14f** in 89% yield. Finally, the scope of the transformation could be extended outside PROTACs to the synthesis of other chemical biology tools. For example, biotinylation of oxaprozin was successfully achieved, forming **14g** in 32% yield.

Time is a critical factor during any medicinal chemistry campaign, where speed is one of the most valuable resources. As such, there is a high demand for novel synthetic strategies that can provide efficient access to otherwise synthetically challenging target compounds[49]. In this context, and to better evaluate the enabling potential of this late

stage C–H functionalisation chemistry, we compared the preparation of compounds **14a**–**14g** achieved here in a single step, with their de novo synthesis using traditional routes. Resource-intensive and lengthy (5–14 steps) synthetic sequences would be required to prepare the same derivatives using traditional methods (see the Supplementary Information, Section 8 for proposed de novo syntheses). In contrast, this late stage C–H amidation protocol offers the potential to rapidly form such valuable drug conjugates, while allowing access to underexplored chemical space.

With access to these fully elaborated constructs, we next measured key physicochemical, pharmacological and pharmacokinetic properties of derivatives **14a**–**14e** in vitro (Fig. 6b). Pleasingly, nanomolar binding affinity for CRBN was retained in all cases. The synthesised compounds exhibited a broad range of lipophilicity (chromatographic Log $D$ = 1.2–4.3). At the same time, medium to low levels of exposed polar surface area (ePSA < 150 Å² for **14a**–**14c** and **14e**) were preserved, which is desirable for optimal permeability to facilitate oral absorption[11]. High aqueous solubility was also observed in a number of instances (**14a** and **14d**). Finally, low intrinsic clearance (Cl$_{Int}$) in rat hepatocytes was found for atazanavir derivative **14e**, despite the relatively high log$D$. This would suggest the absence of intrinsic metabolic soft spots being introduced upon C–H amidation. It is important to recognise that these single data points are strictly dependant on the specific combination of the different linker, E3- and POI-ligands and thus do not allow general conclusions. However, these results do provide intriguing insights into the potential of this late-stage C–H amidation to rapidly assemble PROTACs within a good property space for initial biological evaluation, without inherently introducing metabolic liabilities[11].

In summary, we have developed a robust ruthenium-catalysed C–H amidation platform for the modular late-stage synthesis of complex heterobifunctional molecules, including PROTAC-like derivatives and other chemical biology tools. The use of high-throughput experimentation allowed the rapid development of mild conditions suitable for application to pharmaceutically relevant substrates. Using readily available dioxazolone reagents, 15 classes of inherent functional groups successfully guided the C–H amidation on a broad range of complex APIs. Together with high selectivity and functional group tolerance, this enabled the late-stage installation of linkers offering different functional handles for subsequent conjugation strategies. Further derivatisation was demonstrated, allowing fluorescent dye conjugation and access to VHL derivatives. Importantly, application to the preparation of CRBN and biotin conjugates was achieved in a single step, significantly shortening the synthesis of these challenging target molecules in comparison to traditional lengthy routes. Furthermore, the diverse products synthesised displayed a broad range of physico-chemical properties, while preserving high binding affinity for the CRBN E3 ligase. Overall this strategy offers high potential to streamline PROTAC discovery, bypassing the need for time- and resource-intensive de novo synthesis of functionalised POI-ligands typically required in the early identification of novel degraders. Finally, we believe that this methodology will also find broad application in synthetic and medicinal chemistry programs, providing efficient and sustainable access to advanced drug conjugates as therapeutics and chemical biology tools.

## Methods

### General procedure for late-stage C–H amidation (conditions A)
On the benchtop, an oven-dried microwave vial was charged with the appropriate LSF substrate (0.25 mmol), dioxazolone reagent (0.25–0.28 mmol, 1.0–1.1 equiv), and pivalic acid (PivOH, 4.60 mg, 0.08 mmol, 30 mol%). The vial was moved into a glovebox under N$_2$ atmosphere, where [Ru(*p*-cymene)Cl$_2$]$_2$ (7.65 mg, 0.01 mmol, 5 mol%) and silver(I) hexafluorophosphate(V) (AgPF$_6$, 12.6 mg, 0.05 mmol, 20 mol%) were added sequentially. The vial was sealed and taken out

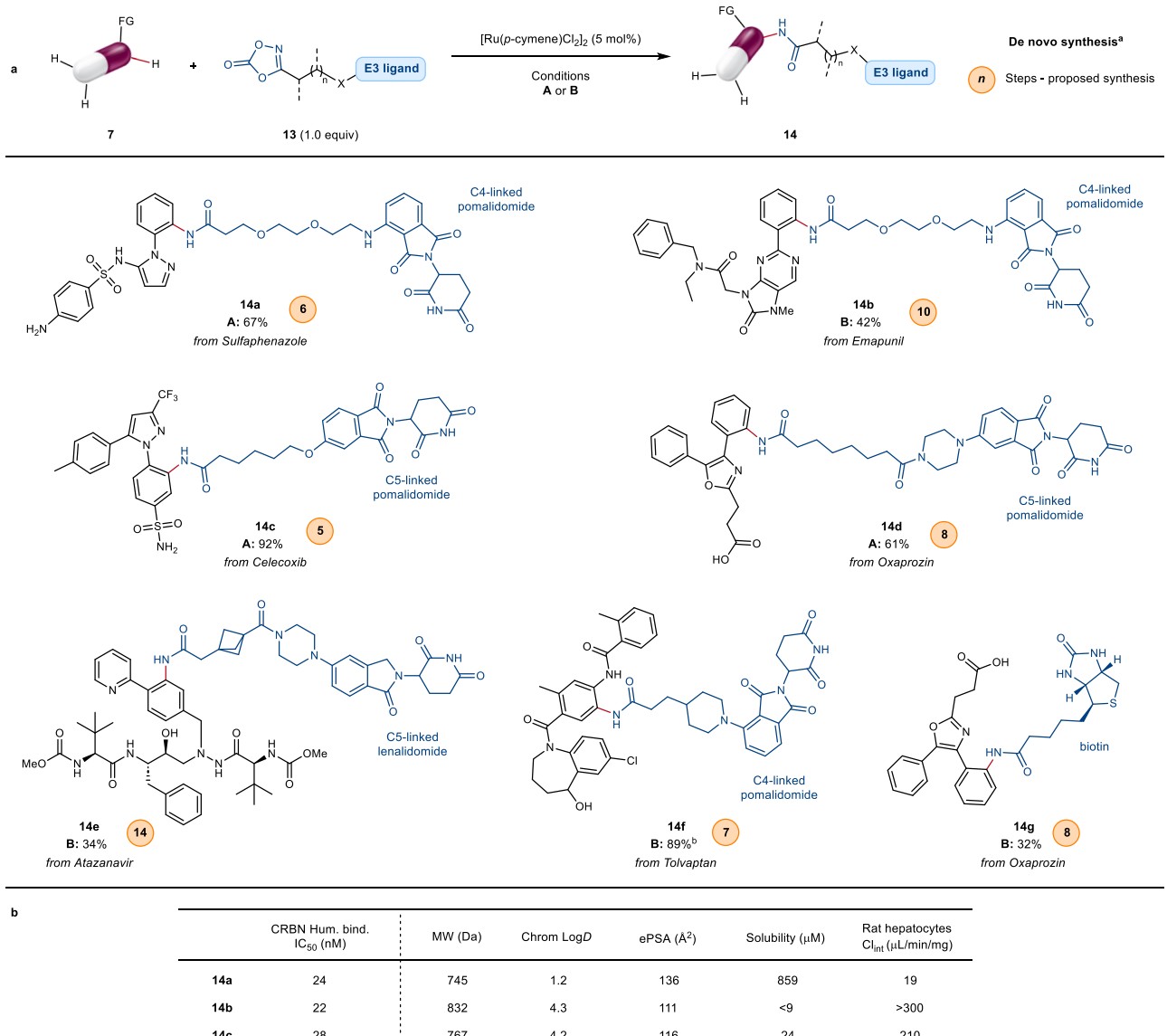

**Fig. 6 | Application of ruthenium-catalysed C–H amidation to the single-step synthesis of heterobifunctional derivatives as PROTACs and chemical biology tools. a** Direct access to fully elaborated CRBN and biotin conjugates via late-stage C–H amidation. Yields refer to the isolated product. Conditions **A**: substrate **7** (0.1–0.25 mmol), dioxazolone **13** (1.0 equiv), [Ru(*p*-cymene)Cl₂]₂ (5 mol%), AgPF₆ (20 mol%), PivOH (30 mol%), TFE (0.1 M), 60 °C, 16 h. Conditions **B**: substrate **7** (0.1–0.25 mmol), dioxazolone **13** (1.0 equiv), [Ru(*p*-cymene)Cl₂]₂ (5 mol%), AgSbF₆ (20 mol%), (PhO)₂PO₂H (30 mol%), DME (0.1 M), 60 °C, 16 h. ᵃ De novo synthesis: number of steps to prepare the depicted product via a traditional lengthy synthesis as opposed to this LSF approach. The number of steps is based on a proposed synthetic route as the compounds are unknown (see the Supplementary Information, Section 8 for more details). ᵇ Pivalic acid (PivOH) was used in place of (PhO)₂PO₂H. **b** Human CRBN binding and physicochemical profile of products **14a–14e** (*n* ≥ 3). FG directing functional group, TFE 2,2,2-trifluoroethanol, DME 1,2-dimethoxyethane, CRBN cereblon E3 ligase, MW molecular weight, chrom Log*D* chromatographic Log*D*, ePSA exposed polar surface area, Cl_int intrinsic clearance in rat hepatocytes.

| | CRBN Hum. bind. IC₅₀ (nM) | MW (Da) | Chrom LogD | ePSA (Å²) | Solubility (μM) | Rat hepatocytes Cl_int (μL/min/mg) |
|---|---|---|---|---|---|---|
| **14a** | 24 | 745 | 1.2 | 136 | 859 | 19 |
| **14b** | 22 | 832 | 4.3 | 111 | <9 | >300 |
| **14c** | 28 | 767 | 4.2 | 116 | 24 | 210 |
| **14d** | 22 | 789 | 1.8 | 160 | 696 | 128 |
| **14e** | 23 | 1182 | 3.6 | 135 | 41 | 6.7 |

of the glovebox. 2,2,2-Trifluoroethanol (TFE, 0.1 M, 2.5 mL) was then added by syringe under N₂ atmosphere, and the vial was heated to 60 °C. After stirring for 16 hours, the reaction mixture was allowed to cool down to room temperature and analysed by LC-MS. The solid material was removed by filtration through a plug of Celite, eluting with EtOAc or EtOAc and MeOH [In cases of poor solubility of the product, this filtration step was not performed]. After removal of the volatiles under reduced pressure, the crude material was either purified by automated flash column chromatography, or dissolved in DMSO (3-5 mL) and purified by preparative reverse phase HPLC. The relevant fractions were collected, combined and concentrated or lyophilised to afford the desired product.

[Note: although the reactions were set up in glovebox under N₂ atmosphere, no significant drop in product yield was observed setting up the reaction under air with no exclusion of moisture.]

**General procedure for late-stage C–H amidation (conditions B)**
On the benchtop, an oven-dried microwave vial was charged with the appropriate LSF substrate (0.25 mmol), dioxazolone reagent (0.25–0.28 mmol, 1.0–1.1 equiv), and diphenyl hydrogen phosphate ((PhO)₂PO₂H, 18.8 mg, 0.08 mmol, 30 mol%). The vial was moved into a glovebox under N₂ atmosphere, where [Ru(*p*-cymene)Cl₂]₂ (7.65 mg, 0.01 mmol, 5 mol%), silver(I) hexafluorostibate(V) (AgSbF₆, 17.2 mg, 0.05 mmol, 20 mol%) and 1,2-dimethoxyethane (DME, 0.1 M, 2.5 mL)

were added sequentially. The vial was sealed, taken out of the glovebox and heated to 60 °C. After stirring for 16 hours, the reaction mixture was allowed to cool down to room temperature and analysed by LC-MS. The solid material was removed by filtration through a plug of Celite, eluting with EtOAc or EtOAc and MeOH [In cases of poor solubility of the product, this filtration step was not performed]. After removal of the volatiles under reduced pressure, the crude material was either purified by automated flash column chromatography, or dissolved in DMSO (3-5 mL) and purified by preparative reverse phase HPLC. The relevant fractions were collected, combined and concentrated or lyophilised to afford the desired product.

[Note: although the reactions were set up in glovebox under $N_2$ atmosphere, no significant drop in product yield was observed setting up the reaction under air with no exclusion of moisture.]

### General software information

NMR data were collected using TopSpin v3 and IconNMR v5, and analysed using MesReNova v14. UPLC-MS data were collected and analysed using MassLynx v4. TIBCO Spotfire v11 was used for data visualisation and heat map generation. Collection and refinement of X-ray diffraction data were performed with CrysAlisPro 1.171.42.35a, Olex2.solve, Olex2, and ShelXL. Mercury v4 and MOE 2022.02 were used to visualise X-ray structures, and model protein surfaces.

### Reporting summary

Further information on research design is available in the Nature Portfolio Reporting Summary linked to this article.

### Data availability

The data generated in this study are provided within the paper and the Supplementary Information file. This includes additional structures; unsuccessful dioxazolones; extended optimization data; full details on the LSF informer library screen; crystallographic data for compound **8r**; analysis of accessible exit vectors (the crystallographic data used are available free of charge in the PDB database, under PDB accession codes: 2AW1, 2AQU, 1HWK, and 3S3G); proposed de novo syntheses; experimental details and characterization data; NMR spectra for novel compounds. Crystallographic data for compound **8r** has been deposited at the Cambridge Crystallographic Data Centre, under deposition number CCDC 2251355.

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

## Acknowledgements

We thank J. Kollback, C. Sanchez, and H. Mobarak for purification support and HRMS data collection. We thank S. Liljenberg for the synthesis of pomalidomide-derived acid substrates. Thanks to R. J. Cox, L. Guillemard, and M. Pettersson for helpful discussions and for proofreading this manuscript. D.A., S.D.F., J.R.J. and M.J.J. acknowledge AstraZeneca and the AstraZeneca Postdoc program for financial support. Generous support by the ERC Advanced Grant (no. 101021358 to L.A.) is gratefully acknowledged.

## Author contributions

D.A., S.D.F., J.R.J., L.A. and M.J.J. conceived the project and designed the experiments. M.J.J. and L.A. directed the project. D.A. performed and analysed the experiments. O.D.P. collected, refined, and analysed the X-ray diffraction data. D.A., S.D.F., J.R.J., O.D.P., L.A. and M.J.J. prepared the manuscript.

## Competing interests

The authors declare no competing interests.
