## [Peer Review File · Nature Communications]

REVIEWER COMMENTS

Reviewer #2 (Remarks to the Author):

In this manuscript, the groups of Ackermann and Johansson developed a methodology for the late-stage C–H amidation reaction by Ru(II)-catalysis using dioxazolone derivatives as coupling partners to access fully elaborated PROTACs and drug conjugates. By combining both industrial and academic perspectives, the study allowed the single-step synthesis of PROTAC and biotin conjugates. In addition, by using THE, this study was well executed and the transformation was functional group tolerant.

Interestingly inherent functional groups successfully guided the C–H amidation on various complex APIs (Fig 6). However, a broader scope of APIs, which did not require the presence of a heteroarene or equivalent, would be a real added value to demonstrate the synthetic utility of the methodology.

For several compounds, lower yields were obtained, which somehow impacted the synthetic added value of the approach. Does a mixture of regioisomers be obtained? Do difunctionalized products have been observed in some cases? Further explanations are needed. In the case of 8u, for which the byproduct S13 was isolated in a higher yield than the expected product, a clear statement should be added in both text and scheme.

Except in the case of 8r for which a X-Ray structure was provided, additional data are needed to ascertain the observed regioselectivity, especially in the case of complex molecules (at least in the supporting information).

Supporting information: IRs need to be added.

Minor point:

For compound 8u (19% isolated yield), the authors mentioned “a synthetically useful yield” and this point should be modified.

In this study, an approach was developed for the modular late-stage synthesis of PROTACs and the late-stage functionalization of other drug conjugates, which might find application in synthetic

and medicinal chemistry programs. However, the novelty of this work is questionable based on the previous report from the authors for the peptide late-stage amidations (eg. Chem 2020, 6, 3428–3439) and the use of a known catalytic system (reaction conditions based on ruthenium catalysis, non-cleavable heteroarenes as DGs). Therefore, the manuscript does not meet the criteria of significance of Nature Communications and should be published in a more specialized journal.

Reviewer #4 (Remarks to the Author):

This peer review concerns only the authors' response to the comments of reviewer 3 only.

This manuscript by Ackermann and Johansson describes ruthenium catalysed synthesis of PROTAC-like molecules by the directed late-stage amidation of aryl C-H bonds with dioxazolones in a variety of different contexts.

Reviewer 3 was particularly concerned with the authors claims that this method is suitable for the synthesis of PROTACs.

I guess the main question here is- at what point does a molecule which is comprised of a E3 ligase binder, some form of linker, and a POI ligand become a proteolysis-targeting chimera? Unless I have missed the data, no molecules synthesised have been demonstrated to effect proteasome-dependent on-target protein degradation and are therefore not true PROTACs. It is evident from the given data that compounds that look like proven PROTACs can be accessed- primarily with CRBN ligands and the newly added VHL ligand- so perhaps the headline here should be 'PROTAC-like molecules rather than 'PROTACs'. I accept that this is a little cumbersome and pure wordplay to be technically correct, so I would leave it to the editor here to decide what is acceptable.

In regard to the authors' response to reviewer 3's comments:

- The inclusion of the VHL ligand is welcome, albeit via a less-pleasing multi-step approach which- ultimately- also highlights the value of the synthetic ability of their method by showcasing downstream modification of functionalised intermediates with valuable groups. A look at the substrate scope suggests that it might be the phenyl-thiazole group that is inhibiting C-H amidation- this might be possible to probe with other proven VHL binders that have different para-substitution on the phenyl group (10.1002/cmdc.202200615).

- The reference to 'CRBN-based PROTAC' when referring to the single-step approach is welcome.

- The inclusion of data regarding C-H amidation at positions relevant to PROTAC development are well done.

- The ability for this method to functionalise existing POI ligands would have been supportive of this method being able to produce PROTACs, but the authors mention that this was not forthcoming. The authors significant comments to this are that they 'aimed at providing a first proof of concept for the late-stage derivatisation of complex "POI-like" molecules'. I appreciate that their method would be best suited to rapidly generating a range of different functionalised unproven POI ligand candidates with structures amenable to this synthetic method prior to assessment of degradation with an appropriate assay, but approach is not described. This relates to my first comment regarding whether a molecule is a PROTAC or not, so more careful wording- particularly in the title- would alleviate these concerns.

- The inclusion of the BODIPY example is welcome and, as mentioned above, highlights the value of the possible downstream derivatisations.

Many thanks to the referees for review of this manuscript. We have addressed the comments below, with our responses in blue font. These have considerably broadened and further strengthened the content of the paper.

Reviewer #2:

In this manuscript, the groups of Ackermann and Johansson developed a methodology for the late-stage C–H amidation reaction by Ru(II)-catalysis using dioxazolone derivatives as coupling partners to access fully elaborated PROTACs and drug conjugates. By combining both industrial and academic perspectives, the study allowed the single-step synthesis of PROTAC and biotin conjugates. In addition, by using THE, this study was well executed and the transformation was functional group tolerant.

Interestingly inherent functional groups successfully guided the C–H amidation on various complex APIs (Fig 6). However, a broader scope of APIs, which did not require the presence of a heteroarene or equivalent, would be a real added value to demonstrate the synthetic utility of the methodology.

We thank the referee for the suggestion. The single step conjugation methodology can indeed be extended to complex APIs which do not require heteroarenes directing groups. As shown in figures 4 and 5, a broad variety of directing groups including anilides (products **8m-8s** in Fig 4, **10b**, **10g** and **10i** in Fig 5), benzamides (**8t**, **8u** and **10e**) and ketones (**8v-8y**) were all successful in the reaction with dioxazolones displaying different functionalities and steric requirements.

Furthermore, the possibility to expand the scope of the single-step conjugation was demonstrated with the synthesis of compound **14f**, which leverages an anilide-type directing group. Product **14f** was isolated in 89% yield and added to Figure 6 (page 16).

This change was reflected in the main text on page 5:

“Importantly, type I directing groups were also successful in the single-step conjugation, affording tolvaptan derivative **14f** in 89% yield.”

Furthermore, the numbering of compounds **14g** and **S14** were modified to account for the newly added product **14f**.

b

	CRBN Hum. bind. IC ₅₀ (nM)	MW (Da)	Chrom LogD	ePSA (Å ²)	Solubility (μM)	Rat hepatocytes Cl _{int} (μL/min/mg)
14a	24	745	1.2	136	859	19
14b	22	832	4.3	111	<9	>300
14c	28	767	4.2	116	24	210
14d	22	789	1.8	160	696	128
14e	23	1182	3.6	135	41	6.7

For several compounds, lower yields were obtained, which somehow impacted the synthetic added value of the approach. Does a mixture of regioisomers be obtained? Do difunctionalized products have been observed in some cases? Further explanations are needed.

The LCMS trace of the crude reaction mixture is given for each compound in the Supporting Information, highlighting any byproduct and giving full details on the composition of the crude reaction mixture. To highlight this for the reader, a specific sentence was added to the main text on page 4:

“Furthermore, this led to clean reaction profiles across the scope, with unreacted starting material typically accounting for the remaining mass balance (for LC-MS traces of the crude reaction mixtures see *Supplementary Information, Sections 10.3-10.5*), which in turn enables the high-value substrates to be easily recovered upon purification”

In general, the reaction gave clean reaction profiles, with very high site-selectivity, and difunctionalization was only rarely observed. In most cases with lower yield, unreacted starting material accounted almost exclusively for the remaining mass balance. Comments on this are included throughout the text:

Page 4: "... this led to clean reaction profiles across the scope, with unreacted starting material typically accounting for the remaining mass balance, which in turn enables the high-value substrates to be easily recovered upon purification."

Caption Fig 4: "A broad range of polar, unprotected or sensitive functional groups (highlighted in purple) were tolerated, typically resulting in high mass recovery (see *Supplementary Information, section 10.3* for substrate specific details)."

In some cases, where specific byproducts were identified, details were also provided in the text:

Page 4: "Competing acylation of the primary sulphonamide in celecoxib was observed under conditions B"

In the case of 8u, for which the byproduct S13 was isolated in a higher yield than the expected product, a clear statement should be added in both text and scheme.

This was added in the main text on page 4:

"Despite partial -OH acylation was observed for paclitaxel (22% yield, **S14** in the *Supplementary Information*), it is remarkable that C–H amidation still occurred to form **8u** in 19% yield, enabling rapid access to analogues of this highly complex antineoplastic agent."

And in the caption of Figure 4 (footnote c):

"^c Product **S14**, derived from -OH acylation of paclitaxel was also isolated in 22% yield, while unreacted starting material accounted for the remaining mass balance (see *Supplementary Information, Section 10.3*)."

Except in the case of 8r for which a X-Ray structure was provided, additional data are needed to ascertain the observed regioselectivity, especially in the case of complex molecules (at least in the supporting information).

In all cases, the structural assignment is based on 2D NMR experiments, typically including COSY, HSQC and HMBC, as well as NOESY and ROESY where appropriate, in addition to standard 1H and 13C NMR measurements. These, together with the X-ray structure of compound 8r, were all consistent with the structural assignment and site-selectivity indicated in the manuscript.

In most cases multiplicity along with chemical shift is sufficient to determine site-selectivity. For more complex structures, we have also added additional spectroscopic data to the SI, to further support our structural assignment.

Supporting information: IRs need to be added.

We understand the referee comment, but IRs are something that we don't routinely do. Looking at other synthesis methodology papers in Nature Communications this does not seem to be common. In particular, due to the high structural complexity of the vast majority of our products, IR spectra would fall short in identifying characteristic FGs in these already densely functionalized compounds and would not provide additional information useful for structural assignments.

Minor point:

For compound 8u (19% isolated yield), the authors mentioned "a synthetically useful yield" and this point should be modified.

As mentioned in the point above, this sentence was modified on page 4:

“Despite partial -OH acylation was observed for paclitaxel (22% yield, **S14** in the *Supplementary Information*), it is remarkable that C–H amidation still occurred to form **8u** in 19% yield, enabling rapid access to analogues of this highly complex antineoplastic agent.”

In this study, an approach was developed for the modular late-stage synthesis of PROTACs and the late-stage functionalization of other drug conjugates, which might find application in synthetic and medicinal chemistry programs. However, the novelty of this work is questionable based on the previous report from the authors for the peptide late-stage amidations (eg. Chem 2020, 6, 3428–3439) and the use of a known catalytic system (reaction conditions based on ruthenium catalysis, non-cleavable heteroarenes as DGs). Therefore, the manuscript does not meet the criteria of significance of Nature Communications and should be published in a more specialized journal.

While our previous work - focused on the late-stage diversification of peptides via C–H amidation (Chem 2020, 6, 3428–3439, included as reference 41) - is also characterized by an excellent functional group tolerance, it utilised rhodium catalysis. Now, we have developed conditions based on ruthenium catalysis which we found to be superior to rhodium in enabling the late-stage functionalisation of a broad range of complex pharmaceuticals, while providing a significant reduction in catalyst cost. Ruthenium is approx. 9 times cheaper than rhodium, with a cost of 465 USD/oz and 4200 USD/oz respectively (data retrieved from <https://eibprices.basf-catalystsmetals.com/mp/> on 05/10/2023). Furthermore, the previous rhodium-catalysed diversification is limited to unnatural tryptophan residues bearing a Lewis basic heteroarene directing group at the N1 position. In striking contrast, our current work achieves optimal reactivity with >15 different classes of inherent directing groups, including more weakly Lewis basic functionalities such as amides and ketones. Not only this significantly expands the scope of the transformation, but no additional steps are required to install and remove specialised directing group on the late-stage substrates. As such, we believe that our current work represents a significant advance compared to our prior efforts.

Reviewer #4:

This peer review concerns only the authors' response to the comments of reviewer 3 only.

This manuscript by Ackermann and Johansson describes ruthenium catalysed synthesis of PROTAC-like molecules by the directed late-stage amidation of aryl C-H bonds with dioxazolones in a variety of different contexts.

Reviewer 3 was particularly concerned with the authors claims that this method is suitable for the synthesis of PROTACs.

I guess the main question here is- at what point does a molecule which is comprised of a E3 ligase binder, some form of linker, and a POI ligand become a proteolysis-targeting chimera? Unless I have missed the data, no molecules synthesised have been demonstrated to effect proteasome-dependent on-target protein degradation and are therefore not true PROTACs. It is evident from the given data that compounds that look like proven PROTACs can be accessed- primarily with CRBN ligands and the newly added VHL ligand- so perhaps the headline here should be 'PROTAC-like molecules rather than 'PROTACs'. I accept that this is a little cumbersome and pure wordplay to be technically correct, so I would leave it to the editor here to decide what is acceptable.

We thank the referee for raising this point on the use of the term PROTAC. While the molecules we synthesised have a PROTAC-like chemical structure, we did not collect in-vitro degradation data for these compounds, and we realise the term PROTAC or degrader might be questionable. To address this point, we modified the title and the main text:

Title:

“Late-stage synthesis of heterobifunctional molecules for PROTAC applications via ruthenium-catalysed C–H amidation”

Text:

We now refer to heterobifunctional compounds, PROTAC-like molecules or generally conjugates throughout the text.

In regard to the authors' response to reviewer 3's comments:

- The inclusion of the VHL ligand is welcome, albeit via a less-pleasing multi-step approach which- ultimately- also highlights the value of the synthetic ability of their method by showcasing downstream modification of functionalised intermediates with valuable groups. A look at the substrate scope suggests that it might be the phenyl-thiazole group that is inhibiting C-H amidation- this might be possible to probe with other proven VHL binders that have different para-substitution on the phenyl group (10.1002/cmdc.202200615).

We thank the referee for the suggestion, and this is indeed the case. Addition of 1 equivalent of simple 5-phenyl thiazole to the standard reaction conditions was found to inhibit the transformation. This is indicated now in reference 57:

“Addition of 1 equiv of N-Ac VHL ligand to the standard reaction conditions completely inhibited C–H amidation of both substrates **1** and **2**. This is likely due to poor tolerance towards the thiazole moiety. Indeed, only traces of products **5** and **6** were found upon addition of 5-phenylthiazole (1 equiv) to otherwise standard reaction conditions.”

Future efforts in our lab will focus on the use of other VHL binders that have different para-substitution on the phenyl group.

- The reference to 'CRBN-based PROTAC' when referring to the single-step approach is welcome.
- The inclusion of data regarding C-H amidation at positions relevant to PROTAC development are well done.
- The ability for this method to functionalise existing POI ligands would have been supportive of this method being able to produce PROTACs, but the authors mention that this was not forthcoming. The authors significant comments to this are that they 'aimed at providing a first proof of concept for the late-stage derivatisation of complex "POI-like" molecules'. I appreciate that their method would be best suited to rapidly generating a range of different functionalised unproven POI ligand candidates with structures amenable to this synthetic method prior to assessment of degradation with an appropriate assay, but approach is not described. This relates to my first comment regarding whether a molecule is a PROTAC or not, so more careful wording- particularly in the title- would alleviate these concerns.
- The inclusion of the BODIPY example is welcome and, as mentioned above, highlights the value of the possible downstream derivatisations.